# Ezrin and Its Phosphorylated Thr567 Form Are Key Regulators of Human Extravillous Trophoblast Motility and Invasion

**DOI:** 10.3390/cells12050711

**Published:** 2023-02-23

**Authors:** Maral E. A. Tabrizi, Janesh K. Gupta, Stephane R. Gross

**Affiliations:** 1School of Life and Health Sciences, Aston University, Birmingham B4 7ET, UK; 2Institute of Metabolism and Systems Research, University of Birmingham, Birmingham B15 2TT, UK; 3Fetal Medicine Centre, Birmingham Women’s NHS Foundation Trust, Birmingham B15 2TT, UK

**Keywords:** ezrin, motility, trophoblasts, migration, invasion, cytoskeleton, placenta

## Abstract

The protein ezrin has been shown to enhance cancer cell motility and invasion leading to malignant behaviours in solid tumours, but a similar regulatory function in the early physiological reproduction state is, however, much less clear. We speculated that ezrin may play a key role in promoting first-trimester extravillous trophoblast (EVT) migration/invasion. Ezrin, as well as its Thr567 phosphorylation, were found in all trophoblasts studied, whether primary cells or lines. Interestingly, the proteins were seen in a distinct cellular localisation in long, extended protrusions in specific regions of cells. Loss-of-function experiments were carried out in EVT HTR8/SVneo and Swan71, as well as primary cells, using either ezrin siRNAs or the phosphorylation Thr567 inhibitor NSC668394, resulting in significant reductions in both cell motility and cellular invasion, albeit with differences between the cells used. Our analysis further demonstrated that an increase in focal adhesion was, in part, able to explain some of the molecular mechanisms involved. Data collected using human placental sections and protein lysates further showed that ezrin expression was significantly higher during the early stage of placentation and, importantly, clearly seen in the EVT anchoring columns, further supporting the potential role of ezrin in regulating migration and invasion in vivo.

## 1. Introduction

During the early stage of development, trophoblast invasion into the decidualised endometrium is a key regulator to establish the precursor of the placenta, the first step of implantation. During this phase, the controlled invasion and migration of extravillous trophoblast cells (EVT) into the maternal decidua is a vital aspect of placental development. Shallow invasion has been linked to poor supplies of both blood and nutrients to the developing foetus, whilst excessive invasive trophoblast cells lead to the loss of the normal plane of cleavage from the uterine wall and massive haemorrhage at delivery. Both of these conditions have been shown to ultimately lead to pregnancy complications such as placenta accreta, preeclampsia, and foetal growth restriction. A factor whose expression has been linked to foetal growth restriction is the ezrin protein [1,2]. Ezrin is a member of the ERM (ezrin, radixin, moesin) protein family that has been shown to function as an important linker protein between F-actin filaments in the cellular cortical layers and membrane-associated proteins on the cell surface [3,4]. Among different functions, ezrin has been shown to be a key regulator of plasma membrane activities such as cell shape [4] and cell surface structure [5], as well as cell adhesion [6] and cellular migration/invasion [4]. Significant work has now established the molecular signature of the ezrin protein and demonstrated how the different domains found within are responsible for some of its functions. An N-terminal four-point-one, ezrin, radixin, moesin (FERM) region facilitates the interaction, either directly or indirectly, of ezrin with transmembrane domains, and in doing so, allows for anchoring to the plasma membrane. The C-terminal region of ezrin interacts with cortical F-actin [7]. The unmodified full length ezrin protein is usually found in a dormant configuration and not bound to actin [8] due to the masking of the actin-binding domain at the C-terminus by its own N-terminal FERM region [9,10]. One the other hand, the phosphorylation of the residue threonine Thr567 and/or the addition of phosphatidylinositol bisphosphate [11,12,13] leads to conformational changes that uncover the C-terminus region and therefore promote binding to cortical F-actin. The interaction of ezrin with F-actin has been significantly studied in the cancer setting, where ezrin is thought to facilitate cancer cell motility and invasion [14]. Elevated levels of ezrin are considered to be an important step towards carcinogenesis with increased levels seen in prostate and pancreatic cancer, to highlight just a few [15,16,17]. There is also a clear link between high expression and the metastasis of numerous solid tumours [18,19,20].

A role for ezrin in the reproduction field is, however, much less characterised. Ezrin has been suspected to play a role in blastocyte adhesion and has been shown to be distinctively expressed in the primitive endoderm as well as the trophectoderm cell surface [21]. It has been shown in placental microvilli of human [22] and rat origin [23], where it is a major component of actin cytoskeletal structures. Ezrin is the most abundant of the ERM proteins and has been reported to be expressed in the apical membrane of syncytiotrophoblasts in the microvillous membranes [23,24]. It was initially discovered and characterised in choriocarcinoma trophoblast Jeg-3 cells [25] but also found in BeWo cells [26] as well as in extravillous and villous trophoblasts [27]. The roles and functions of ezrin in trophoblast and placental development are, however, not clear, especially in the early stages of gestation. Significant similarities have been highlighted between the physiological process of trophoblast invasion during placental implantation and the pathological effects of cancer invasion [28,29]. The functions of proteins known as metastasis-inducing proteins (MIP) such as S100P have been liked to trophoblast migration [30]. Given the importance of ezrin expression and Thr567 phosphorylation as key regulators of cellular motility and cancer metastasis, we sought to determine whether ezrin could equally contribute to cellular motility and invasion in the context of EVT cells. In this study, we show that reducing either ezrin protein levels and/or its phosphorylated activated form, through the use of siRNA delivery and/or specific inhibitors, respectively, leads to a significant reduction in EVT lines and primary cells’ abilities to both migrate and invade with significant changes made to key motile markers of their cytoskeletal organisation. We further show that levels of ezrin are significantly increased in the early stages of human placental implantation and its expression can be clearly seen in EVT anchoring columns, indicating that this protein may be a key regulator of the physiological process of trophoblast implantation, offering a yet-unexplored role for ezrin in non-diseased states.

## 2. Materials and Methods

### 2.1. Human Placental Tissues

All the works using placental samples were performed in accordance with the ethical principles for medical research outlined in the Declaration of Helsinki 1964 and per subsequent revisions (https://www.wma.net/, accessed on 5 June 2018). Samples of placenta tissues obtained either after the elective termination of pregnancy from the first trimester (*n* = 3) or second trimester of gestation, (*n* = 3) as well as after delivery (*n* = 4) were collected with approval by the National Research Ethics Service (NRES) Committee North West—Haydock (study approval number 13/NE/2005). Samples were fixed in 4% (*w*/*v*) paraformaldehyde and embedded in paraffin wax prior to further processing. Samples of placenta tissues for EVTs isolation were obtained immediately after the elective termination of pregnancy from the first trimester of gestation (8–12 weeks). Placental samples were collected with the approval of the Health Research Authority—West Midlands, Edgbaston Research Ethics Committee (NHS REC 15/WM/0284 and AHRIC REF 1245-SG). All samples and tissues were collected in accordance with relevant guidelines and regulations and written informed consent was obtained from all women recruited into the study.

### 2.2. Cell Lines and Culture

Cells lines used were characterised as first-trimester extravillous trophoblasts EVT HTR8/SVneo, Swan71, and SGHPL4/SGHPL5, which were all kind gifts from Prof. Charles Graham (Queen’s University, Canada), Prof. Gil Mor (Yale University, USA) and Prof. Guy St J. Whitley (St George’s, University of London, UK), respectively. Jeg3 cells and BeWo are well-characterised choriocarcinoma cells which were obtained from Dr Emmanuel Karteris (University of Brunel, UK). Cells were cultured in their respective media, as previously described [30,31,32,33]. NSC668394 was purchased from Merck (UK) and prepared/used as previously described [34].

### 2.3. EVT Purification

The protocol for isolating EVTs was adapted as previously described [35]. Briefly, first-trimester placental samples were gently washed in Ham’s F12 (Sigma, Welwyn Garden City, UK). Chorionic villi were physically separated and digested in 0.25% (*w*/*v*) trypsin solution (Sigma, UK). Placental tissue was filtered and diluted in 25% (*v*/*v*) FBS in Ham’s F12 medium (Sigma, UK) before centrifugation for 5 min at 450× *g*. The supernatant was discarded and the pellets were pooled and resuspended before being layered over a Pancoll solution (Pan Biotech, Aidenbach, Germany. Density 1.077 g/mL). Samples were spun at 750× *g* for 20 min and the EVT band was aspirated and collected into clean tubes prior to a final centrifugation at 500× *g*. The resulting EVTs were resuspended in trophoblast complete medium (TCM; Ham’s F12 without Phenol red, 20% (*v*/*v*) FBS, 100 units and 0.1 mg/mL penicillin/streptomycin, 2 mM L-Glutamine), seeded onto 35 mm tissue culture dishes, and left to settle overnight before changing the medium to fresh TCM.

### 2.4. siRNA Ezrin and Control Delivery

Cells seeded in 24-well plates were grown to 25–50% confluency before being transfected with double-stranded siRNA (Qiagen, Manchester, UK) for ezrin (siRNA 7: SI02664228) and siRNA 9 (SI04384170)) or with a mock control siRNA (SI03650318) in OptiMEM (Gibco, Paisley, UK) and normal medium using INTERFERin transfection reagents (Polyplus, Illkirch-Graffenstaden, France). The concentration of siRNA used throughout the experiments was 5 nM. Cells were left in the presence of the different siRNAs for 72 h prior to collection for Western blotting analysis. For motility/invasion and immunostaining, cells were left to grow for 48 h prior to starting the analysis.

### 2.5. Western Blotting

Cell lysate samples were collected by either scrapping or homogenisation, respectively, in 1× PBS with protease inhibitors prior to sonication and dilution in 5× Laemmli buffer and equal loading (15 μg), then separated onto 10% (*w*/*v*) polyacrylamide gels and transferred to PVDF membranes with 80 mA per blot for 2 h prior to blocking in blocking buffer (3% (*w*/*v*) BSA in PBS). Ezrin/phospho Thr567-ezrin, α-tubulin antibodies (all from Abcam, UK; See Appendix A), as well as moesin and radixin (Insight Biotech, Wembley, UK; See Appendix A), were diluted in blocking buffer and incubated overnight at 4 °C, prior to washing and incubation with the relevant secondary antibodies conjugated to HRP and ECL development (anti-mouse or anti-rabbit, Dako and Sigma, Ely, UK). Original, uncropped, and unadjusted images are available as Appendix A.

### 2.6. Immunofluorescent Staining

Immunofluorescence was carried out as previously described [30,36]. Briefly first-trimester trophoblast HTR8/SVneo, Swan71, or SGHPL4/SGHPL5 as well as BeWo or Jeg-3 cell lines, either untreated or 48 h following treatments, were plated at a concentration of 15 × 10^3^ cells/well onto fibronectin-coated (2.5 μg/cm^2^) glass coverslips in a 24-well plate. The cells were washed once in cytoskeleton buffer (CB: 150 mM NaCl, 5 mM MgCl_2_, 5 mM EGTA, 5 mM glucose, 10 mM 2-(*N*-morpholino)ethanesulfonic acid, pH 6.1) and fixed with 3.7% (*w*/*v*) paraformaldehyde in CB at 37 °C for 20 min, followed by permeabilisation with 0.1% (*v*/*v*) Triton X-100 in CB for 10 min. Blocking solution (5% (*v*/*v*) goat serum in CB) was added and incubated for 60 min. Primary antibodies against ezrin/phospho Thr567-ezrin (Abcam, Cambridge, UK) and paxillin (Invitrogen, Paisley, UK) (Appendix A) were incubated for 45 min at 37 °C. After washing three times with blocking solution, the cells were incubated with the appropriate secondary anti-rabbit or anti- mouse antibodies labelled with FITC or TRITC (Dako, Ely, UK), respectively, in blocking solution for 45 min at 37 °C. For actin staining, rhodamine phalloidin (Invitrogen, Paisley, UK) was also added with secondary antibodies at a concentration of 0.6 μm. After washing with blocking solution, coverslips were rinsed once with water and mounted in Vectashield mounting medium (Vector Labs, Peterborough, UK) before being viewed using an Epifluorescence Leica DMI400B microscope. Regions of specific cellular localisation were magnified 13× to highlight distinctive morphologies. Focal adhesion numbers as well as cell numbers were counted on each image to quantify the average focal adhesion count per cell.

### 2.7. Immunohistochemistry

Immunohistochemical staining for ezrin and counterstaining were performed on human placental tissues, as previously described [30]. Tissue sections were deparaffinised, and heat-induced antigen-retrieval was performed in citrate buffer (pH 6.0) using a pressure cooker (Prestige Medical, Blackburn, UK). Non-specific protein binding was blocked by incubation with 10% (*v*/*v*) normal goat serum (Vector Labs, UK) for 1 h. Primary antibody against ezrin (Abcam, UK), anti-HLA-G antibody (Abcam, UK), or an anti cytokeratin7 antibody (Leica Biosystems, Newcastle Upon Tyne, UK) were carried out at 4 °C overnight (Appendix A). After washing in 0.1% (*v*/*v*) Tween 20 (Sigma-Aldrich, Gillingham, UK) in Tris-buffered Saline pH 7.4, sections were then incubated with the appropriate second antibody conjugated to horseradish peroxidase (HRP). The staining was analysed by using a Nikon inverted microscope and Image-J analysis software. The quantification was performed using the Optical Density Calibration—ImageJ software 1.52a (https://imagej.nih.gov/ij/docs/examples/calibration/, accessed on 4 September 2022). First, optical density (OD) was calibrated on FIJI-Image J, and then the images were analysed by selecting the following options: Image, Colour, Colour Deconvolution, and H DAB. Then, “Colour_2”, which shows the brown DAB staining, and therefore is the equivalent of the expression of the protein of interest, was used to measure the OD. The desired areas in each image were selected through Analyse, Tools, ROI manager and by manually choosing the areas with tissues. The OD in each image was measured by selecting Analyse and then Measure. Finally, the average ODs from each trimester were compared with each other. To quantify the ratio of ezrin-positive pixels to total pixels in each image, Colour Deconvolution and H DAB were selected in Image J. Then, the threshold was automatically adjusted on “Colour_2” (DAB), and the DAB-positive pixels along with total pixels per image were measured.

### 2.8. Motility/Invasion Assay

Motility and invasion abilities were measured using Boyden polycarbonate transwell membranes, as described previously [30]. Following siRNA treatment for 48 h or NSC668394 and serum deprivation by growing the cells in 0.5% (*v*/*v*) FBS-containing medium for a further 24 h, 25 × 10^4^ or 50 × 10^4^ cells, for Swan71 and HTR8/SVneo, respectively, were seeded in 0.5% (*v*/*v*) FBS-containing medium on 8 μm polycarbonate transwell membranes (Greiner, Stonehouse, UK) without (motility) or with congealed Matrigel (invasion; Sigma, UK) against a gradient of 5% or 10% (*v*/*v*) FBS medium in the outer wells for HTR8/SVneo and Swan71, respectively. Cells were left to migrate through the membrane for 24 h prior to fixing and staining using a Diffquik histochemical kit (Reagena, Toivala, Finland) following the manufacturer’s instructions. The stained cells on the lower side of the membrane were counted using a light microscope with a 20× objective lens, selecting 5 random fields. Data for this experiment are presented as the mean values ± S.E. relative to controls (percentage) from 4 replicate wells for each set of conditions.

### 2.9. Trypan Blue Exclusion/MTT Conversion

To measure the cell growth rate, 15 × 10^3^ HTR8/SVneo and Swan71 cells were seeded in 24-well plates along with the desired treatments, as described earlier, and left to grow for 24–72 h. For the trypan blue exclusion, wells were washed once with PBS and trypsinised prior to being counted using a haemocytometer. MTT in PBS (Thermo Fisher Scientific, Oxford, UK) was added into each well for 1 h prior to cells being washed in PBS and solubilised in DMSO. Absorbance was measured at 570 nm using a plate reader (BioTek Potton, Peterborough, UK). Data for this experiment are presented as the mean values ± S.E. relative to controls (percentage) from 4 replicate wells for each set of conditions.

## 3. Results

### 3.1. Ezrin Is Expressed in Human Extravillous Trophoblasts in Anchoring Columns In Vivo

The high expression of ezrin has been shown during blastocyst activation prior to implantation, being found in the apical domains of the outer cells during blastocyst formation [21,37]. The expression of ezrin in extravillous and villous trophoblasts has also been reported before (mRNA and protein levels) [27], but no information is available, to our knowledge, to detail ezrin’s localisation during gestation, especially in the early stages of trophoblast invasion in the anchoring columns. Human placental sections from the first and second trimesters, as well as at term, were processed using immunohistochemistry staining for ezrin as well as the trophoblast markers cytokeratin 7 and HLA-G (Figure 1). Significant differences in expression were found between the different stages of pregnancy. These changes could be seen at the protein levels either by Western blotting (Figure 1A,B) or after immunohistochemistry (Figure 1C,D and Table 1), resulting in a 45% decrease in the levels of ezrin as gestation progresses to term. Levels of ezrin were significantly reduced throughout the gestational period between the first trimester and second trimester (*p* = 0.0027 for Western blot analysis and *p* = 0.0014 for immunohistochemistry quantification) or at term (*p* < 0.0001 for Western blot analysis and *p* < 0.0001 for immunohistochemistry quantification). Further analysis of the immunohistochemistry was conducted using the percentage of cell positivity quantifications to analyse expression levels in the different trophoblast subtypes (Table 1). As a whole, we found that significant reductions were seen from both the first and second trimesters compared to full term (*p* < 0.001 and *p* < 0.01 respectively). It is interesting to note that these significant changes in ezrin expression throughout gestation were also found across the majority of the different trophoblast subsets, as all cytotrophoblasts, syncytiotrophoblasts, and proximal column extravillous trophoblasts showed a significant reduction in protein levels as the pregnancies progressed (Table 1). At the cellular level, ezrin was found mainly in the cytoplasm of cytotrophoblasts and syncytiotrophoblasts, with the highest levels seen in the apical membranes of the syncytiotrophoblasts (Figure 1C). To further establish if ezrin could also be found in extravillous trophoblasts (EVT) and invasive anchoring columns, samples were analysed using serial sections of placental villi from the first trimester (Figure 1E). HLA-G was used as a prominent marker of EVT and its expression was clearly visible at the tip of the anchoring columns. Interestingly, high levels of ezrin were found in these cells, although, unlike HLA-G, the staining was also observed in cells at the proximal and middle ends of the anchoring columns. Expression was also seen in the proximal syncytiotrophoblasts. Taken together, these results show that the ezrin protein is predominantly expressed in the trophoblast cells, including the extravillous trophoblast subsets and their anchoring columns, and that ezrin levels appear to be highest at times when trophoblasts are the most invasive during the early stages of placental implantation.

### 3.2. Expression and Localisation of Ezrin in Trophoblast Cell Lines

Having shown that ezrin is found to be expressed in EVT cells in human placental sections, we sought to establish whether ezrin could also be seen in cell lines in order to provide a tractable experimental system. Using a panel of lines, including the choriocarcinoma cell lines Jeg-3 and BeWo, and the EVT cells HTR8/SV neo, Swan71, SGHPL4, and SGHPL5 cells, we sought to analyse both ezrin expression levels as well as its subcellular location. Western blot analysis was performed and quantification was assessed (Figure 2A,B). Robust and similar expressions of ezrin were seen in all cells tested. Immunofluorescence was also carried out to seek further information about ezrin’s cellular localisation (Figure 2C). In the Jeg-3 cells, ezrin was found mainly in the cytoplasmic and pericellular organisation, whilst the EVT-like cells demonstrated a very specific localisation with an abundant presence of ezrin in protrusions. These extensions were particularly extensive in number and length in the HTR8/SVneo and Swan71 cell background (Table 2). Ezrin activation and its ability to bind both to membrane and cytoskeletal structures are regulated by conformational changes. Intramolecular interactions in the C-terminal domains mask the actin-binding sites [10], but the phosphorylation of a conserved threonine residue at position 567 releases this inhibition and results in the relocalisation of the protein to the actin-rich membrane extensions [38]. To determine the extent of Thr567 phosphorylation in the different trophoblast cell lines, we measured its levels and its cellular localisation. A significant difference in the levels of phosphorylated Thr567 ezrin was seen across the different cells, with HTR8/SVneo trophoblasts demonstrating high levels of the phosphorylated proteins, whilst the Swan71 cells, in contrast, offered much lower levels (Figure 2A). It is also interesting to note that beyond the different levels of Thr567 phospho-ezrin in the trophoblast cell lines, there were some equally clear changes in the migration patterns of the different proteins, suggesting, most likely, changes in post-translational modifications, including phosphorylation at Thr567 in these cell populations (Figure 2A and Appendix A). Choriocarcinoma Jeg-3 cells showed important levels of phosphorylated Thr567 ezrin, and these proteins were seen as intense clusters in the cytoplasm and pericellular space, as well as defined fibrillary foci at the membrane periphery (Figure 2D). EVT-like cells HTR8/SVneo and SGPHL4 demonstrated the localisation of phosphorylated Thr567 ezrin either in the nucleus or in very defined cellular microvilli structures at the cell periphery, most of them colocalising with the actin cytoskeleton (Figure 2D). Interestingly, staining in the Swan71 cells was similar to that of the other EVT-like cells, but much weaker across, still presenting defined protrusion structures, which were also more difficult to observe, and confirming the low phosphorylated Thr567 ezrin levels seen (Figure 2A). These data demonstrate that endogenous levels of ezrin, as well as its phosphorylated form, can be detected in all trophoblast cell lines and can therefore be used as models to silence ezrin expression.

Given that HTR8/SV neo and Swan71 cells are some of the best-characterised EVT models and given the significant differences between the levels or localisation of both ezrin and its phosphorylated Thr567 state in these two cells, we decided to use both lines for further studies. 

### 3.3. Regulating the Expression and Activity Levels of Ezrin in EVT Cell Lines

To study the potential role of ezrin in EVT trophoblast behaviours, we first sought to establish strategies to successfully regulate the levels and/or activity of the protein. For the former, targeted siRNA delivery was used. Cells were treated with different ezrin-targeted siRNAs or their mock control counterpart for 72 h prior to cell lysate collection and Western blotting analysis in both EVT HTR8/SVneo and Swan71 lines (Figure 3A–D). The addition of specific siRNAs was found to significantly reduce the levels of ezrin expression (Figure 3A–D). Treatment with siRNA7 reduced the ezrin concentration in cells by more than 40% (*p* < 0.05) and 75% (*p* < 0.0001) in HTR8/SVneo and Swan71 cells, respectively. siRNA9 led to the lowering of protein expression by 60% (*p* < 0.0001) and 80% (*p* < 0.0001) in HTR8/SVneo and Swan71 cells, respectively. Given the high sequence homology between ezrin and other ERM proteins and the possible cross-reactivities of antibodies, it was also important to identify that our analysis affected specifically ezrin levels only. Both moesin and radixin were found to be expressed in the HTR8 and Swan71 trophoblast cells (Figure 3A–D) but with some cell-specific differences, as a double band could be visible for moesin in the Swan71 trophoblast cell extracts. Moesin has been shown to be post-translationally modified and the presence of different bands has been shown in the context of ERM proteins in numerous cells lines and tissues [39,40,41,42], and it is unclear whether these bands are the result of post-translational modifications or the cross-reactivity of the antibodies. Treatment with either siRNA7 or 9 did not lead to any significant changes in the levels of radixin or moesin (*p* > 0.05 for all conditions) in our experiments, further demonstrating the specificity of both the knock-downs and our ability to detect ezrin levels in relation to other known ERM factors. To determine the effects of the ezrin loss-of-function on these cells, we also set about to reduce Thr567 phosphorylation using inhibitors. NSC668394 has been shown to interact specifically with ezrin and inhibits both its phosphorylation as well as actin binding [34]. To verify that this inhibitor reduced Thr567 phosphorylation without affecting the overall levels of ezrin, HTR8/SVneo and Swan71 cells were incubated with 5 μm NSC668394 for 24 h prior to either the collection of cell lysate for Western blotting (Figure 3E,G) or fixation for immunostaining (Figure 3I). Phosphorylated Thr567 ezrin pools were found to be significantly reduced in HTR8/SVneo cells but not in Swan71, with levels lowered by more than 80% in the former (*p* < 0.001) and remaining close to the untreated ones in the Swan71 cells ((*p* > 0.05; Figure 3E–H). Across all the conditions, the concentrations of total ezrin were found to be unaffected after treatments, indicating that the inhibitor does not actually affect the overall levels of expression (*p* > 0.05). To gain further insights regarding the cellular localisation of the different ezrin forms, mock or NSC668394-treated HTR8/SVneo and Swan71 cells were stained for either form of the protein (Figure 3I). The localisation of the proteins was seen throughout the cytoplasm but also present at the cell periphery, in specific protrusions. Treatment with the NSC668394 inhibitor resulted in a general reduction in these protrusions with a significant reduction in their numbers in both the ezrin and Thr567-phosphorylated experiments in the HTR8/SVneo background (70% *p* < 0.001) but also in the Swan71 cells, albeit to a much lesser effect (15% reduction *p* < 0.05), as determined after the quantification of the number of protrusions per cells (Appendix A), whilst the overall levels of total cellular ezrin were not affected. Altogether, this analysis shows that ezrin protein levels and potential activities can be significantly and specifically affected by siRNAs and inhibitor delivery and that the two EVT-like cells offer differential responses in relation to the treatment carried out.

### 3.4. The Knock-Down or Inactivation of Ezrin Results in Significant Reductions in EVT-like Cell Motility

Ezrin has been shown to be a key regulator of cellular migration in multiple human cancer lines [17,43,44,45,46,47]. Its activation via Thr567 phosphorylation has also been reported to be key in this process for both healthy and diseased-state cells [4,48,49]. We therefore sought to determine whether ezrin or its potential Thr567-phosphorylated form can similarly promote EVT cell migration. Cell motility was monitored using Boyden chamber assays after ezrin knock-down by siRNA delivery or treatment with the NSC668394 inhibitor in the HTR8/SVneo and Swan71 trophoblast cells (Figure 4A,D or Figure 4B,E, respectively). Mock-treated samples demonstrated no significant changes in the number of cells able to migrate across the Boyden membranes (*p* > 0.05 for HTR8/SVneo and *p* > 0.05 for Swan71) in the two cell systems used. HTR8/SVneo cells grown in the presence of siRNA7 or siRNA9 presented a significant reduction, by at least 30% (*p* < 0.05) and 60% (*p* < 0.0001), respectively, in their motility (Figure 4A). Similar observations could be made when using the Swan71 cells, as treatment of these cells with the same siRNAs equally resulted in a significant reduction in migratory abilities by about 50% (*p* < 0.0001) and 72% (*p* < 0.0001; Figure 4D). To determine whether phosphorylation of ezrin was also crucial for motility changes, the same trophoblast cells were treated with 5 μm NSC668394 inhibitor for 24 h prior to seeding into Boyden chambers (Figure 4B,E for HTR8/SVneo and Swan71 trophoblast cells, respectively). Incubation with NSC668394 induced a significant 80% reduction (*p* = 0.0001; Figure 4B) in cellular motility in the HTR8/SVneo background. A comparison between the inhibition of the phosphorylation status and the knock-down of the protein was also assessed in relation to cellular motility to determine the levels of inhibition following either NSC668394 or ezrin siRNA treatment (Table 3). Both treatments were seen to significantly reduce cellular motility in the HTR8/SVneo background (all *p* < 0.01) with further reduction seen when treating the cells with NSC668394 (*p* < 0.05 against both siRNAs). This was in stark contrast to the work conducted in the Swan71 cells, where treatments only with siRNAs were shown to result in significant decreases in cell migration. Migratory rates after treatment with NSC668394 were found to be lowered by around 15% in the Swan71 cells (*p* < 0.05; Figure 4E). This result could be explained by the low levels of phosphorylated Thr567 levels seen in these cells (Figure 2A) and the inability of NSC668394 to affect these levels in this line at the concentrations used (Figure 3G–H), further supporting the specificity of this small molecule.

Changes in cellular motility can be associated with the remodelling of cytoskeletal architecture and cellular protrusion, as we have shown for S100P and trophoblasts previously [30]. Furthermore, ezrin has been shown, at least in cancer cells, to regulate the number and size of focal adhesion [46] when affecting their migration/invasion. Here, we now sought to determine if the motility of trophoblast cells might also be affected by ezrin through a similar mechanism. The focal adhesion marker paxillin was used, as it is considered a vital adapter that aggregates into the focal complex at the early stages of their formation [50] and because focal adhesion assembly is a key process regulating cellular motility and the proteins within undergo profound remodelling over time [51]. Its localisation within the focal adhesion, as well as the actin cytoskeleton, was studied in control cells and in their treated counterparts incubated with ezrin siRNAs or NSC668394 (Figure 4C,F), and the total numbers of focal adhesions per cell were quantified (Table 4 and Table 5). siRNA7 and siRNA9 treatments in both HTR8/SVneo and Swan71 cells resulted in significant increases of 45–92% in the number of focal adhesions per cell (Table 4). The biggest increases in their numbers were seen when using siRNA9 and correlated with the reduction in cell motility in both cell types. Treatment with the ezrin inhibitor NSC668394 also demonstrated a significant increase by over 60% in the number of focal adhesions in the HTR8/SVneo background (*p* < 0.0001; Table 5). All in all, these data show that affecting either the activation or the level of ezrin in trophoblast cells leads to significant changes in EVT cell motility and that these changes correlate with the number of paxillin-containing focal adhesions.

### 3.5. The Knock-Down or Inactivation of Ezrin Results in Significant Reductions in EVT Cell Invasion

Ezrin has also received a large amount of interest in relation to its ability to promote cancer cell invasion and metastasis [17,47,52] with the inhibition of its phosphorylation at Thr567 being seen as one avenue to prevent its activation and reduce cancer cell invasiveness as well as potential metastasis [34,53]. Having shown that ezrin regulated trophoblast cell motility, we set about to study its potential role in promoting the invasion of the HTR8/SVneo and Swan71 cells. As before, loss-of-function experiments by the knock-down of the expression of ezrin via siRNA or the inhibition of its phosphorylated Thr567 status through the addition of NSC668394 were carried out prior to Boyden chamber Matrigel invasion (Figure 5). Invasion was seen to be inhibited with either siRNA used in all trophoblast cells and resulted in a reduction of 30% (*p* < 0.05) and 65% (*p* = 0.0001) with siRNA7 and siRNA9 in HTR8/SVneo, respectively, and 50% (*p* = 0.01) and 60% (*p* = 0.0001) with siRNA7 and siRNA9 in Swan71, respectively. These numbers were similar to those observed with the migration experiments. Incubation with the NSC668394 compound led to different responses with the cells used. Whilst its presence resulted in a significant reduction of more than 70% of the invasive properties of the HTR8/SVneo cells (*p* < 0.01), the same concentration was found to insignificantly change (*p* > 0.05) the invasion of Swan71 cells, again mirroring data obtained in the migration experiments of the same cells.

### 3.6. The Knock-Down or Inactivation of Ezrin Does Not Lead to Changes in Cell Viability and Proliferation

We have shown that modulating ezrin expression affects both the motility and invasion of trophoblast cells. The expression of ezrin has been linked to cell proliferation, at least in the cancer state [43,54], but, to date, there is no evidence of this protein regulating such processes in trophoblasts. To determine whether the effects reported so far in HTR8/SVneo and Swan71 are linked to increases in cell viability, we measured cell growth over 48 h after either ezrin levels had been knocked down or after NSC668394’s addition. The incubations used were well within the time frame used to measure invasion and migration (Figure 6A,B for HTR8/SVneo and Figure 6C,D for Swan71). Incubation with either of the siRNAs resulted in no significant changes in the cell numbers when compared to either untreated or mock-treated samples for both the HTR8/SVneo (*p* > 0.05) or Swan71 cells (*p* > 0.05). Similarly, the addition of the NSC668394 inhibitor did not affect the proliferation of the cells over the course of the experiment in all conditions tested (HTR8/SVneo (*p* > 0.05) or Swan71 cells (*p* > 0.05)), indicating that cell viability was probably not compromised in our experiments, as the cells were proliferating normally over the course of the experiment, with a doubling of the population every 24 h indicating that the defects seen in motility are not due to a reduction in cell numbers.

### 3.7. Inhibiting Ezrin Phosphorylation in Primary Extravillous Trophoblasts Results in a Significant Reduction in Motility and Invasion

Our data so far show that the expression and activation of the ezrin protein differentially regulate both the migratory and invasion rates of trophoblast cell lines. To determine whether similar observations could also be observed in primary trophoblast cells, first-trimester human EVT cells were isolated and studied (Figure 7). Staining for these cells using the HLA-G marker demonstrated that more than 80% of the cells were differentiated EVT cells after immunofluorescence. High levels of the ezrin proteins were seen in these cells (Figure 7A), and its localisation was found to be similar to that observed in the different lines, with an extensive organisation in cellular protrusions at the cell periphery (Figure 7B). Interestingly, phosphorylated Thr567 ezrin was found to be present at high levels, as demonstrated by the significant intensities obtained after Western blot analysis, especially when compared to other trophoblast cell lines (Appendix A), whilst its localisation mirrored that of the total ezrin pools, with a large amount detected in numerous cellular protrusions along with the cortical actin network (Figure 7C). Both the levels and localisation seen in primary EVTs mirrored data obtained with the HTR8/SVneo cell lines (Figure 2). To establish whether we could inhibit the levels of phosphorylated Thr567 ezrin, EVT primary cells were treated with the NSC668394 inhibitor for 24 h prior to analysing the changes by staining (Figure 7C,D). The addition of the NSC668394 inhibitor resulted in a significant reduction of more than 50% in the overall phosphorylated Thr567 signal (*p*< 0.001). The same concentration of the small molecule led to a significant reduction of about 30% (*p* < 0.05) and 40% (*p*< 0.01) in their rates of migration and invasion, respectively (Figure 7E,F). These data demonstrate that primary human extravillous trophoblasts express ezrin and that this protein is actively phosphorylated at residue Thr567 and that inhibiting such post-translational modifications leads to a significant reduction in their motility and invasion.

## 4. Discussion

Elevated levels of ezrin are considered to be an important step towards carcinogenesis, with increased levels seen in numerous cancers and its expression linked to the metastasis of numerous solid tumours [18,19,20]. A role for ezrin in the reproduction field is, however, much less characterised and its functions in trophoblast and placental development in the early stages of gestation are not clear. Because significant similarities have been highlighted between the physiological process of trophoblast invasion during placental implantation and the pathological effects of cancer invasion [28,29], functions of proteins known as metastasis-inducing proteins (MIP), such as S100P, have been liked to trophoblast migration [30]. We initially sought to establish whether ezrin is expressed in the early stages of placental development and whether it could be found in human extravillous trophoblasts and within their anchoring/invading columns. To achieve this, human placental sections (and protein samples) at differential gestation periods were used.

Significant levels of ezrin were found in the cytoplasm of cytotrophoblasts and syncytiotrophoblasts, with the highest levels seen in the apical membrane of the syncytiotrophoblasts (Figure 1), similar to what has been reported previously [55,56]. This is not surprising, as ezrin has been shown to be a major component of placental microvilli [57]. It is thought that the protein plays some key role in regulating the overall microvillus formation at least in epithelial cells [58]. Microvilli in the placental syncytium effectively sense and interact with the fluid environment [59], and recent work has shown that these microvilli regulate the transfer of material between the foetal and maternal blood flow [26]. In this work, BeWo and villous trophoblast cells were found to form abundant microvilli of varying lengths where ezrin was relocalised to the apical membrane, further suggesting a role for ezrin to facilitate these exchanges. Interesting and novel was the fact that we could also detect ezrin in the invasive trophoblast columns, localised within the cytoplasmic and apical membrane of the EVTs (Figure 1). To determine if there were changes in the expression of ezrin in trophoblasts over the course of gestation, we stained and quantified the levels of expression for the whole tissues as well as within the different trophoblast populations. We found that the highest level of ezrin was seen during the first and second trimesters of gestation in all trophoblast subtypes analysed, while the expression was significantly reduced at term (Figure 1). This suggests a potential involvement of the protein in earlier stages of gestation. This observation correlates with prior studies, where placental ezrin expression was shown to decrease in late pregnancy in rats [23]. Moreover, the involvement of ezrin and the two other members of the ERM family in implantation was recommended previously by another study, as they detected higher expression of ERM proteins in implantation-competent blastocysts compared with dormant blastocysts in mice [21]. It is, however, important to highlight that another piece of work [27] suggested an inverse expression pattern when high levels of ezrin protein were shown to offer a stronger signal in the later stage of gestation, but no quantification of the immunohistochemistry analysis was provided.

We also wanted to characterise the expression levels of ezrin in trophoblast cells and used a collection of background lines as well as first-trimester isolated primary cells. Whilst extensive work has been done to document its expression in the choriocarcinoma Jeg-3 linage, there is, to our knowledge, no report that has characterised ezrin in EVT cells. We found that, in all cases, ezrin protein levels were relatively consistent in all trophoblast cells used. There were, however, significant differences in ezrin subcellular localisation and its Thr567-phosphorylated state in the different cell systems (Figure 2 and Figure 7 and Appendix A). Whilst ezrin offered a cytoplasmic and microvilli-like organisation in choriocarcinoma lines (BeWo and Jeg-3), some pools were found to be specifically localised near or at the plasma membrane in long cellular protrusions in all EVT cells, including primary cultures. This observation was even more pronounced when looking at the phosphorylated Thr567 version of the protein. Levels of phosphorylation were found to be different whereby HTR8/SV neo and primary cells demonstrated very high levels compared to other EVT cells, with Swan71 cells presenting very low levels of phosphor-Thr567 ezrin. To our knowledge, it is the first time that ezrin has been shown in such a specific cellular location in extravillous trophoblasts. Although we are aware that other data using Jeg-3 trophoblast cells have also indicated the organisation of ezrin into microvilli [60] or in trophoblast giant cells [61]. Ezrin’s presence in cellular protrusion has been evidenced in transfected MCF7 and observed by immunoelectron microscopy [62] as well as in keratinocytes and breast cancer cells using immunofluorescence [63,64]. Furthermore, ezrin has been reported to be a component of actin-rich structures such as focal adhesion and filopodia, lamellipodia, and membrane ruffles [65]. Structures found in EVT cells were, however, very different in terms of number and length. Interestingly, our data suggest that the Thr567-phosphorylated ezrin pools were enriched in the cellular protrusions. This was clearly observed after the use of the NSC668394 inhibitor and the fact that whilst overall ezrin localisation and intensity were not affected following such treatments, there were important reductions in the presence of phosphorylated-Thr567 ezrin in these protrusions in the HTR8/SVneo and primary extravillous trophoblast backgrounds (Figure 3 and Figure 7). This is, again, in line with the presence of the phosphorylated active form present in microvilli [60] and other observations that have shown that the localisation of the proteins might indicate the level of protein activation. It is known that the active (open) ezrin, similar to other ERM family members, links the cytoskeleton to the plasma membrane via PIP2, and therefore is localised within the cell cortex, while the inactive (closed) ezrin is located in the cytoplasm [8].

We found that ezrin and its phosphorylated form were present in cellular protrusions in specific locations of the extravillous trophoblast cells, whether lines or primary. Given that these extensions are finger-like cytoplasmic structures containing F-actin filaments and that they extend well beyond the leading edge, we speculate that these structures are filopodia. Numerous migrating cells have been shown to display filopodia, including trophoblast cells [66]. Whilst our data suggest the importance of ezrin and its phosphorylation for the formation of these filopodia, the mechanisms as to how these are regulated are not known. One of the most likely scenarios is that ezrin and its phosphorylation may regulate Rho activity. Ezrin has been shown to regulate members of the Rho family [67], including Rho guanine nucleotide exchange factors, Rho GTPase-activating proteins, and Rho GDP-dissociation inhibitors. Given the wealth of work that has been carried out to link filopodia formation and extension being regulated by the Rho family through actin filament polymerisation, there are multiple factors within this extensive family which may be controlled by ezrin and lead to the changes seen.

There is a large amount of evidence that has shown that ezrin expression is associated with enhanced cellular motility and invasion, as well as metastasis and poor prognosis for cancer patients [17,68,69]. Consequently, we aimed to see how this protein regulates trophoblast motility and invasion through loss-of-function experiments via ezrin knock-down or the inhibition of Thr567 phosphorylation. For the former, two different sequences of siRNA against ezrin were presented (siRNA 7 and 9), as they were the most effective in knocking down ezrin in HTR8/SVneo and Swan71 cells (Figure 3) and resulted in a significant reduction in motility (Figure 4) and invasion (Figure 5). This is in line with current work using siRNA delivery to monitor motile and invasive phenotypes in cancer cells [17,70,71]. This is also the first time that ezrin has been directly linked to changes in migration and invasion in extravillous trophoblasts, although we are aware that the roles of this factor in the choriocarcinoma Jeg-3 linage were recently reported [72].

Thr567 ezrin phosphorylation levels were seen to be significantly different across the trophoblast cells studied. Both HTR8/SVneo and primary extravillous trophoblast cells demonstrated high levels confirming the suitability of this cell line as a good model of the first-trimester trophoblast line. Interestingly Swan71, as well as the SGHPL4 and SGHPL5 background, presented much lower levels, although the localisation of the phosphorylated form was seen in the cellular protrusion, presumably resulting in its activation and its phenotypical relocation to key structures, such as that shown for the promotion of apical microvilli formation [73,74]. Ezrin is activated through phosphorylation at thr567 by different kinases. Members of the PKC serine/threonine kinase family such as PKCα, Ι, and γ [75] and redundant kinases lymphocyte-oriented kinase (LOK, STK10) and sterile 20-like kinase (SLK, Ste-20) [76] phosphorylate ezrin at Thr567. Given that both PKC [77] and LOK/SLK [78] have been found in trophoblasts, we decided to use the cell-permeable quinoline NSC668394, which inhibits ezrin Thr567 phosphorylation primarily via its binding to ezrin and not through the inhibition of kinase activity [34]. Interestingly, NSC668394 treatment was found to lead to a significant reduction in both the Thr567-phosphorylated status and motility/invasion, in the HTR8/SVneo background as well as in primary extravillous trophoblasts, but had little effect in the Swan71 cells. This, again, may be explained by the generally very low levels of phosphorylation seen in the latter. Differential responses in relation to ezrin phosphorylation in cells following incubation with the NSC668394 inhibitors have been reported, for instance in cancer states such as myeloid leukaemia cell lines, where different cells treated with the same dose resulted in differential levels of phosphor-ezrin, especially at concentrations below 12.5 μm [79]. All in all, these data indicate that both ezrin and its phosphorylated form play important roles in motility and invasion in line with work carried out in cancer cells where the relocalisation of ezrin in membrane and microvilli also improved these properties [80].

During the promotion of cellular migration, rapid formation and the turnover of the FAs during directional cell movement pulls the cell body forward, while their rapid turnover at the rear end detaches them from the extracellular matrix. We therefore sought to determine if the ezrin-induced cell motility reported in extravillous trophoblasts involved observable FA changes by analysing the localisation pattern of paxillin, an FA component, upon ezrin knock-down or after inhibiting Thr567 phosphorylation. Reducing ezrin expression in both HTR8/SVneo and Swan71 cells resulted in a significant increase in the number and size of the paxillin-containing focal adhesion, in line with previous work showing a reduction in FA dynamics in breast cancer for the proper disassembly and turnover of FA and invadopodia structures [46]. The role of ezrin in this cellular signalling is not well characterised, especially in EVT cells, but a possible candidate could be the protein calpain, which is yet another factor that has been linked, albeit indirectly, to the promotion of focal adhesion disassembly through cleavages of specific targets [81]. Ezrin is a calpain substrate [82,83,84] and has been shown to positively regulate calpain activity.. Ezrin has also been shown to interact with the focal adhesion kinase (FAK [6]) and regulates its activation. Given that both calpain and FAK expressions have been found in trophoblasts [85], including the extravillous HTR8/SVneo line [86,87], these are some of the potential mechanisms by which ezrin could regulate the changes in focal adhesion seen in our work. Whilst the work carried out in this report concentrated on ezrin and its phosphorylated Thr567 residue, we are also aware that there are other phosphorylated sites on the protein that could explain the significant changes in separation on gels (Figure 2 and Appendix A), such as Tyr477 [88,89], Ser66 [90], Thr235 [91], and Tyr145 [92]. Tyr353 is also one of the important amino acid residues that can be phosphorylated and is involved in the PI3-kinase/Akt signalling [93]. It is possible that these and other post-translational modifications may also be associated with ezrin’s activation. It is therefore important to consider that studying the role of these other regions of ezrin is essential to offer a better understanding of how ezrin regulates the cytoskeletal changes in trophoblasts and its potential role in early placenta development.

## Figures and Tables

**Figure 1 cells-12-00711-f001:**
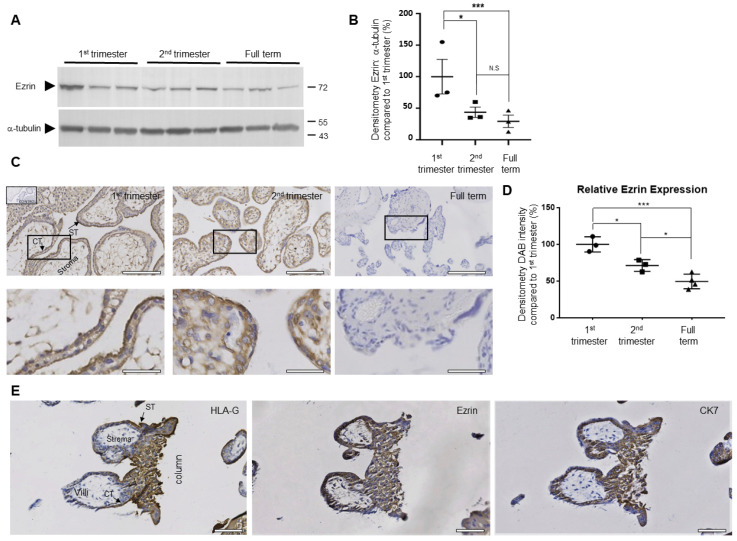
Ezrin is expressed in trophoblasts, including extravillous trophoblasts and anchoring columns during the first trimester in human placental tissues. Expressions of ezrin proteins were analysed on both lysates (**A**,**B**) and paraffin-embedded placental sample sections (**C**,**E**) obtained from different gestational periods (first trimester (*n* = 3), second trimester (*n* = 3), or full term (*n* = 4)). Protein lysates of equal loading were separated by SDS-PAGE electrophoresis and transferred onto PVDF membranes before Western blotting for ezrin or α-tubulin, and cropped images are presented (**A**). Levels of ezrin were measured by densitometry analysis after Western blotting and normalised to α-tubulin for all samples. Data are presented as percentage means ± SEM of 2 independent experiments compared to the first trimester (**B**). Immunohistochemistry staining using either ezrin (**C**) or a panel of trophoblast marker antibodies (HLA-G as well as cytokeratin 7 (CK7)) as well as ezrin (**E**) were also counterstained as described in the methods section. Images for (**C**) show the overall structures of the placental sections with enlarged sections corresponding to the focused regions of the highlighted cells. Bar corresponds to 150 μm in the wide views and 25 μm in the zoomed-in regions. Images of anchoring columns of serial human placental tissues are also presented (**E**). Bar corresponds to 100 μm. Arrows indicate cytotrophoblast cells (CT); syncytium (ST) and stroma are also highlighted (**C**,**E**). The quantification of ezrin DAB staining and intensity in the 1st, 2nd, and full-term sections (**D**). Data of an individual representative experiment are presented as the mean values ± SD of 3 independent samples (**D**). Statistical analysis (**B**,**D**) shows ± SD compared to the first-trimester samples of an individual representative experiment. * *p* < 0.05 and *** *p* < 0.001 (one-way ANOVA); N.S.: not significant.

**Figure 2 cells-12-00711-f002:**
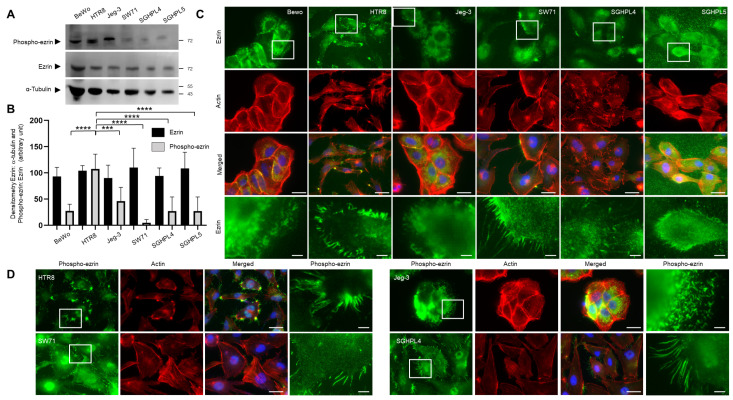
Ezrin is expressed in all trophoblast cell lines. Choriocarcinoma cells BeWo and Jeg-3 cells, along with EVT cell lines HTR8/SVneo, Swan71, SGHPL4, and SGHPL5 cells, were grown for 48 h prior to the collection of the protein lysates for Western blotting (**A**) or cell staining (**C**,**D**). Cells were collected and solubilised in Lammeli buffer and equal loadings were separated by SDS-PAGE electrophoresis. Western blotting was carried out and membranes were probed for ezrin, phosphor-ezrin Thr567, or α-tubulin and cropped images are presented (**A**). The expression levels of ezrin and phospho-ezrin Thr567 were measured by densitometric analysis, normalised to α-tubulin and ezrin, respectively, and presented as arbitrary units and as mean values ± SD of 3 independent samples. *** *p* < 0.001 or **** *p* < 0.0001 (one-way ANOVA) (**B**). For immunostaining, BeWo and Jeg-3 cells and EVT-like cell lines HTR8 SV/Neo, Swan71, SGHPL4, and SGHPL5 cells were seeded on fibronectin-coated coverslips and grown for 48 h prior to fixation, permeabilisation, and staining for ezrin (**C**), phospho-ezrin Thr567 (**D**), and actin (**C**,**D**). Cells were mounted and viewed using epifluorescence microscopy. Images in the last row correspond to the enlarged regions of the highlighted cells. Bar corresponds to 25 μm in the wide views and 5 μm in the zoomed-in regions.

**Figure 3 cells-12-00711-f003:**
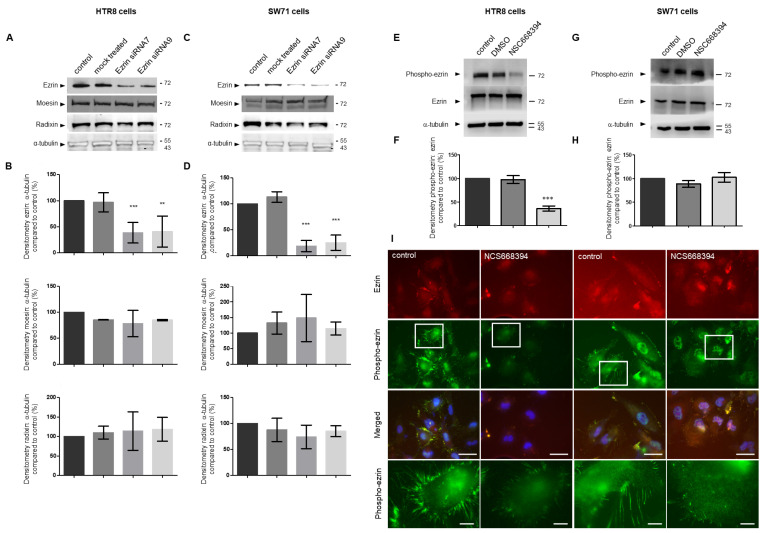
Specific knock-down of ezrin levels but not moesin or radixin or inhibitions of ezrin Thr567 phosphorylation in HTR8/SVneo and Swan71 trophoblastic cell lines. HTR8/SVneo and Swan71 cells were incubated in the presence of different ezrin siRNAs or control siRNAs for 72 h (**A**–**D**) or treated with NSC668394 inhibitors for 24 h (**E**–**I**) prior to collection for protein Western blotting (**A**,**C**,**E**,**G**) or immunofluorescence (**I**). For the determination of protein levels, cells were collected and solubilised in Lammeli buffer and equal loadings were separated by SDS-PAGE electrophoresis prior to PVDF transfer and Western blotting for ezrin, moesin, radixin or α-tubulin (**A**,**C**) or phosphor ezrin Thr567 (**E**,**G**), and cropped images are presented. The expression levels of ezrin, moesin and radixin or Thr567-phosphorylated ezrin and the effects of siRNAs or inhibitors, respectively, were measured by densitometric analysis, normalised to α-tubulin or ezrin and presented as percentage mean values ± SD of 3 independent samples of a representative experiment compared to non-treated control samples. ** *p* < 0.01 or *** *p* < 0.001 (one-way ANOVA). For all proteins, loading orders are the same as for the blots above (**B**,**D**,**F**,**H**). EVT-like lines HTR8/SVneo and Swan71 cells were seeded on fibronectin-coated coverslips and grown for 48 h prior to fixation, permeabilisation, and staining for ezrin and phospho-ezrin Thr567 (**I**). Cells were mounted and viewed using epifluorescence microscopy. Images in the last row correspond to the enlarged regions of the highlighted cells. Bar corresponds to 25 μm in the wide views and 5 μm in the zoomed-in regions.

**Figure 4 cells-12-00711-f004:**
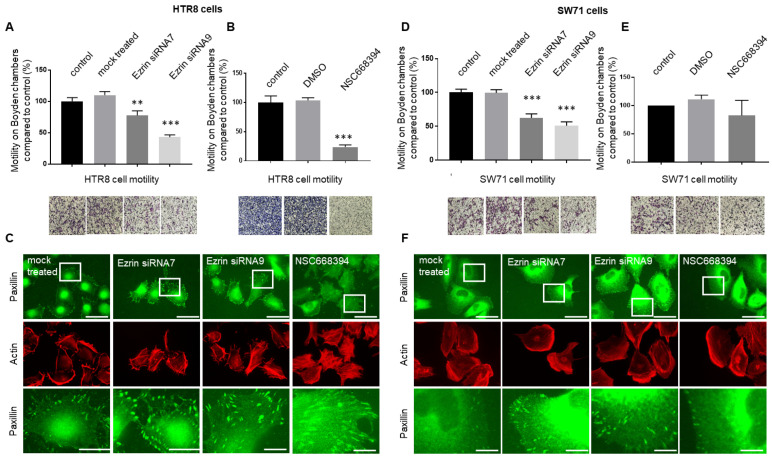
Reduced ezrin levels or its Thr567phosphorylation results in differential reductions in the cellular motility of HTR8/SVneo and Swan71 trophoblasts. HTR8/SVneo and Swan71 cells were incubated in the presence of different ezrin siRNAs or control siRNAs for 72 h (**A**,**D**) or treated with NSC668394 inhibitors for 24 h (**B**,**E**) prior to starvation with low-serum-containing medium. After 24 h, cells were seeded into Boyden chambers for 16 h prior to fixation and staining using the Diffquik histochemical kit for labelling of both nuclei and cytoplasm (**A**,**B**,**D**,**E**). Five random fields were quantified for each chamber. Data are presented as means ± SEM of 4 independent experiments relative to controls (percentage) from 4 replicate wells for each set of conditions. ** *p* < 0.01 or *** *p* < 0.001 compared to control and mock treatments (one-way ANOVA). Images of representative fields of motility/invasion assays were taken with the EVOS XL Cell Imaging System at x20 magnification. Bar corresponds to 100 μm (**A**,**B**,**D**,**E**). For immunofluorescence after siRNA delivery for 48 h, incubation cells were seeded on fibronectin-coated coverslips and grown for a further 48 h before fixing. For the ezrin phosphorylation experiment, NSC668394 inhibitors were added for 24 h prior to fixation, permeabilisation, and staining for paxillin and actin. Cells were mounted and viewed using epifluorescence microscopy (**C**,**F**). Images on the last row correspond to the enlarged regions of the highlighted cells. Bar corresponds to 50 μm in the wide views and 10 μm in the zoomed-in regions.

**Figure 5 cells-12-00711-f005:**
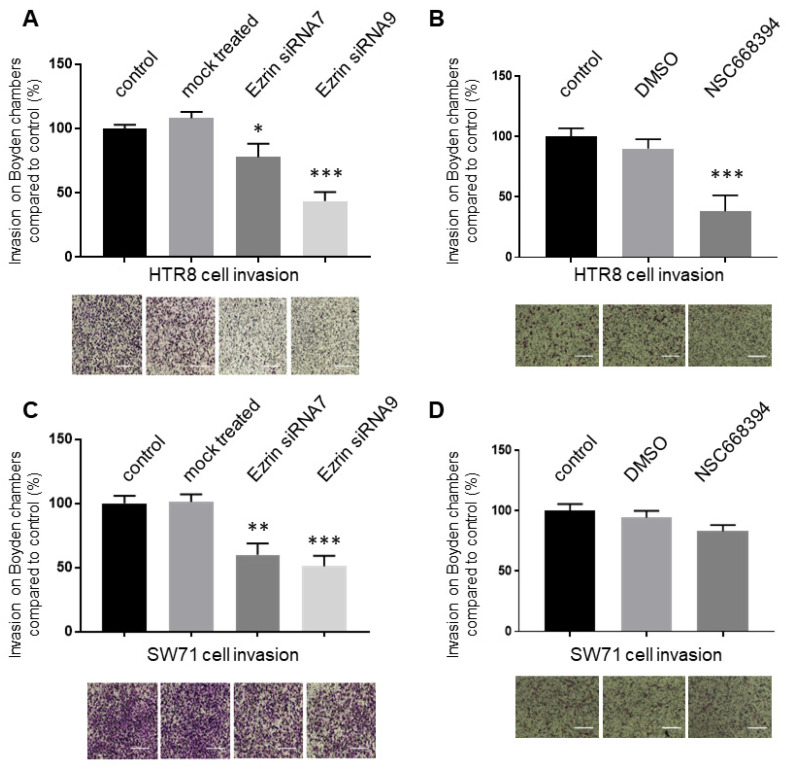
Reduced ezrin levels or its Thr567phosphorylation results in the differentially reduced cellular invasion of HTR8/SVneo and Swan71 trophoblasts. HTR8/SVneo and Swan71 cells were incubated in the presence of different ezrin siRNAs or control siRNAs for 72 h (**A**,**C**) or treated with NSC668394 inhibitors for 24 h (**B**,**D**) prior to starvation with low-serum-containing medium. After 24 h, cells were seeded into Matrigel-coated Boyden chambers for 16 h prior to fixation and staining using the Diffquik histochemical kit for the labelling of both nuclei and cytoplasm (**A**–**D**). Five random fields were quantified for each chamber. Data are presented as the means ± SEM of 4 independent experiments relative to controls (percentage) from 4 replicate wells for each set of conditions. * *p* < 0.05, ** *p* < 0.01 or *** *p* < 0.001 compared to the control and mock treatments (one-way ANOVA). Images of representative fields of motility/invasion assays were taken with the EVOS XL Cell Imaging System at 20× magnification (**A**–**D**). Bar corresponds to 100 μm.

**Figure 6 cells-12-00711-f006:**
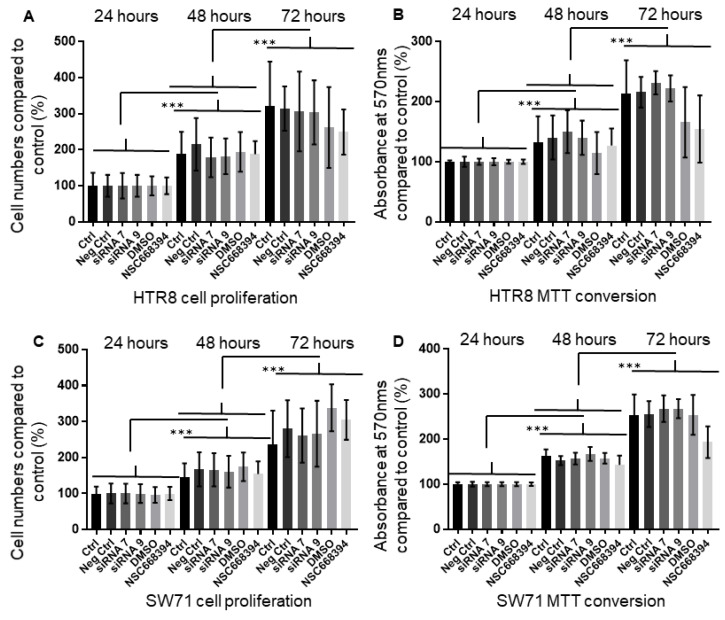
Modulation of ezrin protein levels or Thr567-phosphorylation does not affect HTR8/SVneo and Swan71 trophoblast cell proliferation. HTR8/SVneo and Swan71 cells were incubated in the presence of different ezrin siRNAs or control siRNAs for 72 h or treated with NSC668394 during the course of the experiment (**A**–**D**). Cell lines were seeded into 24-well plates and left to grow for a further 24–72 h before trypan blue exclusion (**A**,**C**) or MTT conversion (**B**,**D**). For the former, cells were trypsinised and removed from the wells prior to counting using trypan blue exclusion. For MTT conversion, MTT was added to the wells and left to be reduced prior to washing cells, and solubilising formazan in DMSO and absorbance measurements were carried out at 570 nms. Data are presented as percentage means ± SD of 3 independent experiments relative to controls from 3 replicate wells for each set of conditions. *** *p* < 0.0001 (one-way ANOVA).

**Figure 7 cells-12-00711-f007:**
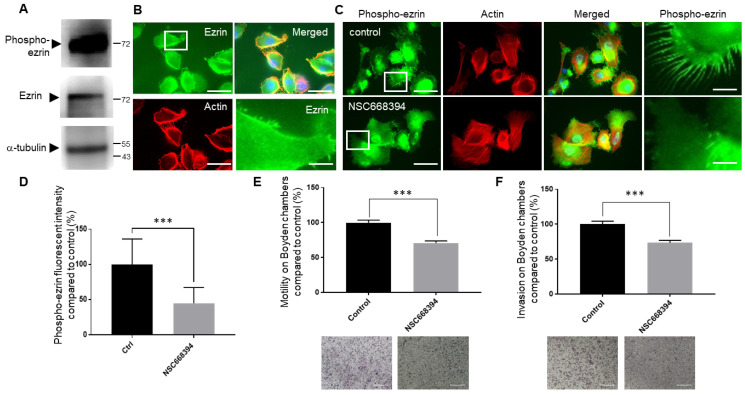
Inhibiting ezrin Thr567phosphorylation results in a significant reduction in both the cellular motility and invasion of human primary extravillous trophoblast cells. Human primary extravillous trophoblasts were grown for 48 h following isolation prior to either the collection of the protein lysates for Western blotting (**A**) or cell staining (**B**,**C**) or seeding for motility (**E**) or invasion (**F**). Cells were collected and solubilised in Lammeli buffer and equal loadings were separated by SDS-PAGE electrophoresis. Western blotting was carried out and membranes probed for ezrin, phosphor-ezrin Thr567, or α-tubulin, and cropped images are presented (**A**). For immunostaining, cells were seeded on fibronectin-coated coverslips and grown for 24 h prior to NSC668394 treatment. Cells were left to grow for a further 24 h prior to fixation, permeabilisation, and staining for ezrin (**B**), phospho-ezrin Thr567 (**C**), and actin (**B**,**C**). Cells were mounted and viewed using epifluorescence microscopy. Images in the last row correspond to the focused regions of the highlighted cells. Bar corresponds to 50 μm in the wide views and 10 μm in the zoomed-in regions. Expression levels of Thr567-phosphorylated ezrin in human primary extravillous trophoblasts treated with NSC668394 inhibitors for 24 h by immunofluorescence (**D**) and presented as intensity percentage mean values ± SD of 3 independent samples of a representative experiment compared to non-treated control samples. *** *p* < 0.001 (one-way ANOVA). For motility and invasion, cells were starved with low-serum-containing medium. After 24 h, cells were seeded into Boyden chambers with or without Matrigel and NSC668394 treatment for 16 h prior to fixation and staining using the Diffquik histochemical kit for the labelling of both nuclei and cytoplasm (**E**,**F**). Five random fields were quantified for each chamber. Data are presented as the means ± SEM of 4 independent experiments relative to controls (percentage) from 4 replicate wells for each set of conditions. *** *p* < 0.001 compared to control and mock treatments (one-way ANOVA).

**Table 1 cells-12-00711-t001:** Ezrin expression reduces throughout the gestation in human placental tissues in the different trophoblast populations.

	Percentage Ezrin Pixel: Total Pixels in Images ± SEM (*n* = 50)	Percentage Ezrin Pixel in STB: Total Pixels in Images ± SEM (*n* = 50)	Percentage Ezrin Pixel in CTB: Total Pixels in Images ± SEM (*n* = 50)	Percentage Ezrin Pixel in pcEVT: Total Pixels in Images ± SEM (*n* = 50)
First Trimester	34.69 ± 3.28	56.24 ± 7.24	81.68 ± 8.72	94.20 ± 5.87
Second Trimester	28.24 ± 2.37	27.24 ± 19.34	55.36 ± 13.30	75.49 ± 10.52
	*p* < 0.05 ^a^	*p* > 0.05 ^a^	*p* < 0.05 ^a^	*p* > 0.05 ^a^
Full Term	12.33 ± 3.41	3.76 ± 5.32	1.52 ± 2.15	29.24 ± 225.53
	*p* < 0.001 ^a^	*p* < 0.01 ^a^	*p* < 0.001 ^a^	*p* < 0.01 ^a^
	*p* < 0.01 ^b^	*p* < 0.05 ^b^	*p* < 0.001 ^b^	*p* < 0.05 ^b^

Expression of ezrin proteins was analysed by pixel quantification after immunohistochemistry staining using the ezrin of paraffin-embedded placental sample sections obtained from different gestational periods (first trimester (*n* = 3), second trimester (*n* = 3), or full term (*n* = 4)). Data of an individual representative experiment are presented as the mean values ± SD of 3 independent samples and correspond to the quantification of DAB-positive pixels against total pixels in each image in first, second, and full-term sections either globally or for specific cell types. STB (syncytiotrophoblast), CTB (cytotrophoblasts), pcEVT (proximal column extravillous trophoblast). ^a^ *p*-value obtained from one-way ANOVA where the ratio of ezrin positive pixels to total pixels in images from first-trimester samples was compared to the second trimester or full term. ^b^ *p*-value obtained from one-way ANOVA where the ratio of ezrin positive pixels to total pixels in images from second-trimester samples was compared to full term.

**Table 2 cells-12-00711-t002:** Inhibiting ezrin phosphorylation with NSC668394 leads to much greater reductions in cellular protrusions in HTR8/SVneo than in Swan71 trophoblast cells.

Cell Lines	Phospho-Ezrin Protrusions Per Cell ± SEM (*n* = 50)	*p*-Value
HTR8/SVneo control	41.00 ± 1.26	
HTR8/SVneo treated with NSC668394	16.41 ± 0.96	*p* < 0.001
Swan71 control	40.81 ± 1.23	
Swan71 treated with NSC668394	35.45 ± 0.91	*p* < 0.05

HTR8/SVneo and Swan71 trophoblast cells, as well as cells treated with ezrin inhibitor NSC668394 for 72 h, were fixed and stained for phospho-ezrin and actin after seeding on fibronectin-coated coverslips. Data shown are means ± SEM corresponding to the average number of phospho-ezrin protrusions observed per cell. The *p*-value was obtained from a one-way ANOVA where the total number of protrusions present in control cells was compared to the inhibitor-treated counterparts.

**Table 3 cells-12-00711-t003:** Comparison of trophoblast motility after inhibiting ezrin using siRNA delivery or NSC668394 treatments.

Cell Lines	Percentage Motility ± SEM (*n* = 50)	*p*-Value ^a^	*p*-Value ^b^
HTR8/SVneo control	100.00 ± 3.46		
HTR8/SVneo treated with ezrin siRNA7	73.92 ± 8.59	*p* < 0.01	*p* < 0.001
HTR8/SVneo treated with ezrin siRNA9	48.40 ± 7.01	*p* < 0.001	*p* < 0.05
HTR8/SVneo treated with NSC668394	24.28 ± 3.91	*p* < 0.001	
Swan71 control	100.98 ± 8.29		
Swan71 treated with ezrin siRNA7	52.17 ± 6.72	*p* < 0.001	*p* < 0.01
Swan71 treated with ezrin siRNA9	39.57 ± 5.59	*p* < 0.001	*p* < 0.01
Swan71 treated with NSC668394	79.88 ± 15.59	*p* > 0.05	

HTR8/SVneo and Swan71 cells were incubated in the presence of different ezrin siRNAs or control siRNAs for 72 h or treated with NSC668394 inhibitors for 24 h prior to starvation with low-serum-containing medium. After 24 h, cells were seeded into Boyden chambers for 16 h prior to fixation and staining using the Diffquik histochemical kit for labelling of both nuclei and cytoplasm. Five random fields were quantified for each chamber. Data are presented as means ± SEM of 4 independent experiments relative to controls (percentage) from 4 replicate wells for each set of conditions. ^a^ *p*-value obtained from one-way ANOVA where the percentages of motile cells in HTR8/SVneo or Swan71 mock control cells were compared to previously ezrin siRNA-treated or NSC668394 counterparts. ^b^ *p*-value obtained from one-way ANOVA where the percentages of motile cells in ezrin siRNA HTR8/SVneo or Swan71 cells were compared to NSC668394 counterparts.

**Table 4 cells-12-00711-t004:** Reduction in ezrin protein levels in trophoblast cells by siRNA delivery significantly increases the number of focal adhesions per cell.

Cell Lines	Percentage Focal Adhesions Per Cell ± SEM (*n* = 50)	*p*-Value ^a^	*p*-Value ^b^
HTR8/SVneo control	100 ± 5.22		
HTR8/SVneo treated with ezrin siRNA7	170.32 ± 11.34	*p* < 0.0001	
HTR8/SVneo treated with ezrin siRNA9	192.48 ± 14.43	*p* < 0.0001	0.0089
Swan71 control	100 ± 4.81		
Swan71 treated with ezrin siRNA7	145.88 ± 10.23	0.0078	
Swan71 treated with ezrin siRNA9	172.78 ± 12.13	*p* < 0.0001	0.1219

HTR8/SVneo and Swan71 mock control cells, as well as cells treated with different ezrin-targeted siRNA for 72 h, were fixed and stained for paxillin and actin after seeding on fibronectin-coated coverslips. Data shown are means ± SEM corresponding to the average number of focal adhesion-containing paxillin observed per cell, presented as a percentage of the control. ^a^ *p*-value obtained from one-way ANOVA where the total number of focal adhesions present in HTR8/SVneo or Swan71 mock control cells were compared to previously ezrin siRNA-treated counterparts. ^b^ *p*-value obtained from one-way ANOVA where the total numbers of focal adhesions present in HTR8/SVneo or Swan71 treated with ezrin siRNA7 were compared to ezrin siRNA9-treated counterparts.

**Table 5 cells-12-00711-t005:** Inhibiting ezrin phosphorylation with NSC668394 in HTR8/SVneo trophoblast cells leads to increases in the number of focal adhesions per cell.

Cell Lines	Percentage Focal Adhesions Per Cell ± SEM (*n* = 50)	*p*-Value
HTR8/SVneo control	100 ± 5.47	
HTR8/SVneo treated with NSC668394	162.11 ± 7.94	*p* < 0.0001

HTR8/SVneo, as well as cells treated with ezrin inhibitor NSC668394 for 72 h, were fixed and stained for paxillin and actin after seeding on fibronectin-coated coverslips. Data shown are means ± SEM corresponding to the average number of focal adhesion-containing paxillin observed per cell, presented as a percentage of the control. *p*-value obtained from one-way ANOVA where the total numbers of focal adhesions present in HTR8/SVneo control cells were compared to the inhibitor-treated counterparts.

## Data Availability

The data presented in this study are available within the article.

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
