# Peer review of "Ezrin and Its Phosphorylated Thr567 Form Are Key Regulators of Human Extravillous Trophoblast Motility and Invasion"

_cells, 2023, doi:10.3390/cells12050711_

Round 1

Reviewer 1 Report

In this paper, authors reported important roles of ezrin and Phospho-ezrin as key regulators of trophoblast motility and invasion. The paper is of some importance for the people in and around this area.  However, I have some major concerns about their results and discussion as shown below. In addition, some sentences are too long (more than seven lines) and redundant for readers in general. I hope authors to make sentences short, compact (concise) in order that readers understand easily.

Major:

1) Authors had better mentione about Declaration of Helsinki in section 2.1. For example, All the works were performed in accordance with the ethical principles for medical research outlined in the Declaration of Helsinki 1964 and per subsequent revisions (https://www.wma.net/).

2) line 225. I cannot agree with author’s report that “Significant differences in the expression signals were found….”. There seems no clear difference between the band density between 1st trimester, 2nd trimester and Full term except for the first lane of 1st trimester. Authors should present results which support their conclusions.    

3) WB pattern of Phospho-ezrin shown in Fig. 2A are not clear. In addition, the molecular mass of Phospho-ezrin in this figure seems to be different among cell lines. The band patterns of Figure 3A are much better than Fig. 2A. Authors should revise this figure.

4) I cannot understand why phosphorylation of ezrin is not inhibited by NSC866394 at all (Fig. 3G). Authors should make a comment of this point.

5) Fig. 3A. I cannot understand why double bands of moesin were observed in SW7 cells (single band in HTR8 cells).   

6) Authors mentioned that “phosphorylated Thr567 ezrin was found to be present at high level…..”. However, authors did not compare the pattern of P-ezrin of primary cells comparatively with that of other cell lines in Figure 7A. Authors should show the pattern of P-ezrin between primary cells and cell lines.

7) Lane 753. Tyr353 is also one of the important amino acid residues for PI3-kinase/Akt signaling (Gautreau, PNAS 96: 7300, 1999).

Minor:

1) There are careless mistakes such as:

Line 627. “migration [30].” Not comma (,).

Line 640. “indication” (not capital letter) is correct.

Line 671. “our data” (not capital letter) is correct.

Authors check sentences more carefully.

Reviewer 2 Report

This manuscript describes the expression and localization of ezrin protein in the placenta throughout gestation. It explains its role in extra-villous trophoblast cell (EVT) migration and invasion using both placental cell-lines and primary cells. The role of this protein is shown by knock-down (with siRNA) and inhibiting ezrin, followed by validation of functional assays, cell migration, and invasion. The manuscript is well-structured, clearly written, and easy to follow. The questions are addressed appropriately with well-designed experiments using placental cell-lines and comparing them with placental primary cells and tissue samples. However, some minor specific concerns are described below.

Major comments:

Authors showed the gestational-age-dependent expression of ezrin in placental tissues throughout pregnancy in the cytotrophoblast (CTB), syncytiotrophoblast (STB), and proximal column extra-villous trophoblast cells (pcEVTs). Later, they mainly focus on the role of ezrin in the migration and invasion of EVT for in-vitro studies. 

Can authors discuss or comment on what might be its role in the cytotrophoblast (CTB) and syncytiotrophoblast (STB)? 

Did you see any difference in the expression pattern of ezrin in CTB, STB, and pcEVT/iEVT in the first-trimester placenta or throughout the gestation by IHC? 

It has been shown that the knock-down of ezrin with siRNA and NSC668394 inhibitor reduces the cellular migration and invasion in both cell-lines. 

Did you test what happens if you over-express ezrin? Does it induce EVT migration and invasion?

Minor comments:

As per the MDPI, Cells guidelines, “data not shown” should be avoided. Please check the guidelines or add these data in the supplementary materials.

 IHC images for Fig,1C are not very clear, specifically 2nd trimester one. 

Round 2

Reviewer 1 Report

I have checked the revised version of the manuscript.  The manuscript has been well brushed although some sentences are still redundant especially their abstract. I have still a major concern about their evaluation of Fig. 2A and B. 

The abstract is redundant. Authors should summarize their abstract in order that readers can understand this research easily. In fact, according to the “Instructions for Authors”, abstract length should be less than 200 words. The present abstract contains 266 words.

Still, I cannot agree with authors’ evaluation of Fig. 2A and B. The densitometry date shown in Fig. 2B does not reflect the WB data shown in Fig. 2A. For example, the expression of p-Ezrin in BeWo cells is underestimated whereas those in SGHPL4 and 5 are overestimated although I can admit that the expression of alpha tubulin in BeWo cells are very high. Authors should normalize the expression of p-Ezrin agaisnt the expression of ezrin (rather than alpha-tublin).

Author Response

We are again very grateful for the time that this reviewer has spent to review our manuscript. We are also thankful to see that our points and responses following on from the first review were all considered and approved. We have made changes, as requested to improve the work, and are pleased to submit a duly revised manuscript and provide a point-by-point response below, detailing how we have addressed the final reviewers’ comments.

For the abstract, we have now reviewed the abstract (LINE 14-37) and made changes in order to shorten sentences and make the delivery clearer. We have also decreased the number of words to 200 in line with Cells requirements.

For the comments in relation to Fig 2A-B, please note that values for the densitometry quantifications for this figure (Figure 2B) were already presented as data either normalised to alpha-tubulin for ezrin or ezrin for phosphor-ezrin, so this has not been changed.

We have now however recalculated the densitometries values and carried out the statistical analysis so that the values are not compared to the BeWo cells and are presented as arbitrary units.

This has not changed the overall distribution of our data and still demonstrate the significant changes in the levels of phospho-ezrin in the HTR8 cells compared to all other lines with the Swan71 cells presenting the lowest levels. We have consequently made no changes to the result sections related to the phospho-ezrin levels as the previous statements are appropriate and correct.

We have however reworded a sentence in the result section LINE 315 “Robust expression of ezrin was seen in all cells…” to reflect the changes since we do not see any variations in overall ezrin in any of our cell lines with this analysis.

Further minor changes have been made throughout the manuscript to fix typos. All changes have been track-changed in the text and some of them were highlighted the sections above.

Finally we once again thank this reviewer as well as the editorial board member for their careful review of our manuscript.